# Scheduling mechanisms to control the spread of COVID-19

**John Augustine**[1], **Khalid Hourani**[2]*, **Anisur Rahaman Molla**[3], **Gopal Pandurangan**[2], **Adi Pasic**[2]

**1** Department of Computer Science and Engineering, Indian Institute of Technology at Madras, Chennai, Tamil Nadu, India, **2** Department of Computer Science, University of Houston, Houston, TX, United States of America, **3** Indian Statistical Institute, Kolkata, India

* kmhourani@khalidsalad.com

**Data Availability Statement:** All source code files are freely available at https://github.com/Covimulation/covimulation.

**Funding:** John Augustine's research is supported in part by an Extra-Mural Research Grant (file

## Abstract

We study scheduling mechanisms that explore the trade-off between containing the spread of COVID-19 and performing in-person activity in organizations. Our mechanisms, referred to as *group scheduling*, are based on partitioning the population *randomly* into groups and scheduling each group on appropriate days with possible gaps (when no one is working and all are quarantined). Each group interacts with no other group and, importantly, any person who is symptomatic in a group is quarantined. We show that our mechanisms effectively trade-off in-person activity for more effective control of the COVID-19 virus spread. In particular, we show that a mechanism which partitions the population into two groups that alternatively work in-person for five days each, flatlines the number of COVID-19 cases quite effectively, while still maintaining in-person activity at 70% of pre-COVID-19 level. Other mechanisms that partitions into two groups with less continuous work days or more spacing or three groups achieve even more aggressive control of the virus at the cost of a somewhat lower in-person activity (about 50%). We demonstrate the efficacy of our mechanisms by theoretical analysis and extensive experimental simulations on various epidemiological models based on real-world data.

## 1 Introduction

The COVID-19 pandemic that is currently sweeping the world has already spread to a large number of people. In many parts of the world it has already infected a significant fraction of the population. For example, in New York City, as early as June 2020, antibody testing suggests that as many as quarter of the population might be infected [1]. However, this is still nowhere near the fraction required for "herd immunity". Many large and medium sized organizations like schools, universities, factories, and businesses have (at least partially) reopened (or soon planning to open) their businesses. In some instances, there have been closures of schools and universities and businesses after reopening due to spike in cases (possibly due to the emergence of new virus strains) leading again to lockdowns and closures and the cycle keeps repeating. Thus one needs effective strategies to reopen and to keep opened organizations

number EMR/2016/003016) funded by the Science and Engineering Research Board, Department of Science and Technology, Government of India and by the VAJRA faculty program of the Government of India. Anisur Rahaman Molla's research supported by DST Inspire Faculty research grant DST/INSPIRE/04/2015/002801. Gopal Pandurangan's research is supported, in part, by NSF grants IIS-1633720, CCF-BSF1717075, CCF-1540512, US-Israel BSF award 2016419, and by the VAJRA faculty program of the Government of India. The funders had no role in study design, data collection and analysis, decision to publish, or preparation of the manuscript.

**Competing interests:** The authors have declared that no competing interests exist.

functioning safely for a longer time, by keeping the virus under check. It is therefore important to study effective non-pharmaceutical intervention mechanisms that can safely reopen human society. Such mechanisms may significantly help in containing, controlling, and slowing the spread of COVID-19, even though it may not fully eliminate it. Moreover, since organizations, cities, and communities have differing levels of disease spread, it is crucial that policymakers weigh in the trade off between containing the spread of the disease versus the impact on productivity.

Our main contribution is the study of a class of intervention mechanisms called *group scheduling*. The inspiration for group scheduling comes from COVID-19 characteristics whereby individuals remain asymptomatic and less infectious for around 4-5 days (the incubation period) from contraction. Subsequently, they either become symptomatic (and can therefore be quarantined) or remain asymptomatic (and still spread the disease). Group scheduling (randomly) partitions the population (e.g. the students in a university or the work force in a company) and schedules them to work *in-person* on different days with possible gaps (i.e., when no group is scheduled). A group is considered quarantined (at home, say) when it is not scheduled to work. Any individual who is symptomatic is quarantined as soon as symptoms are exhibited. Henceforth, when "work" refers to working in-person or in a face-to-face setting. This contrasts with working or performing activities via other mechanisms, e.g., remotely or online.

The key intuition that inspires group scheduling is that individuals can be grouped in a random fashion—reducing the average number of contacts than without grouping—and scheduled to work predominantly with other less infectious individuals. Effective group schedules work in such a way that most infected individuals turn symptomatic (though a significant percentage, about 40%, may remain asymptomatic [2]) and contagious during their break—thereby facilitating quarantining before spreading the infection. Contrary to a full lockdown, our scheduling allows for significant and sustained in-person activity. In fact, we showcase specific group schedules that operate at 70% of a typical five-day (in-person) work week and that can simultaneously dampen the spread of COVID-19 quite effectively.

The main contributions of this paper are summarized as follows:

1. We posit and study group scheduling mechanisms that provide trade-offs between in-person activity and controlling disease spread (these notions are explained in Section 2.1). We show that our mechanisms effectively trade off in-person activity for more effective control of the COVID-19 virus. This interpolates between two extremes: full lockdowns where there is very little in-person activity but with stronger control of the virus verses almost full activity (as in pre-Covid days) but with very little control of the virus.

2. We analyze various mechanisms both theoretically and by simulation. Our theoretical analysis demonstrates the ways in which group scheduling mechanisms help in controlling COVID-19. Our simulations validate the theory and yield insights into the performance of specific mechanisms. Our results show that both the peak number of cases per day as well as the total number of infections can be significantly controlled by following appropriate mechanisms.

3. Our results indicate three specific categories of mechanisms—corresponding to high, medium, and low in-person activity—and their performance in controlling COVID-19 spread. The lower the in-person activity of the mechanism, the higher the control of disease. For specific mechanisms we refer to Ta++ble 3.
   Mechanisms such as (2,3,2) and (3,3,0) achieve even more aggressive control of the virus at the cost of a somewhat lower in-person activity (about 50%); these could be applicable in

situations when the disease spread is more rampant in the population. Depending on the disease spread, one can use an appropriate mechanism that achieves a desired control of the virus at a certain level of in-person activity.

4. A main takeaway from our results is that group scheduling mechanisms help in significantly controlling the spread of COVID-19—reducing the peak number of cases per day as well as the total number of infections. Depending on the rate of infection in a population, different mechanisms are applicable that control the disease spread while maintaining an appropriate level of in-person activity. Our mechanisms provide a basis for safely reopening in-person activity of large organizations such as universities and schools where such mechanisms can be effectively implemented.

5. While our study is specific to COVID-19, our approach and techniques can be modified in a straightforward manner to study the spread of other diseases, including COVID-19 variants (e.g. by modifying the incubation period, the rate of asymptomatic infection).

## 2 Results and discussion

### 2.1 Scheduling mechanisms

We begin with some simple mechanisms that serve as baselines. The *basic mechanism* is one in which there is no intervention or control mechanism of any sort. The disease spreads in the population according to an underlying disease propagation model. (We consider various epidemic models as described in Section 2.2.) Another baseline mechanism is *symptomatic quarantining*, a widely practised mechanism wherein individuals are quarantined if they exhibit any symptom of the disease. As mentioned in Section 2.2.2, we assume that only a certain proportion of infected individuals are symptomatic. According to current CDC estimates [2], about 60% of infected individuals are symptomatic and the rest are asymptomatic. We also consider other percentages to validate the robustness of our results.

We assume a 5 day mean incubation period (in our experiments, we consider a distribution based incubation period model for COVID-19 with mean incubation period of 5 days [3]). Individuals who exhibit symptoms are removed (quarantined) from the group. In this model, an individual will be asymptomatic with probability 0.4 (independently of others) and thus will not be removed until recovery time (assumed to be 14 days after infection); such an individual will remain contagious until they are recovered.

**2.1.1 Proposed group scheduling mechanisms.** We study a family of *group scheduling* mechanisms and highlight specific mechanisms within the family—one, in particular, that epitomizes the optimal trade off between dampening COVID-19 spread and increasing in-person activity. Group scheduling partitions the population into different *randomly* chosen groups and schedules each group on different days with possible gaps between the schedules. A gap is when no group is scheduled. More precisely, a group scheduling mechanism is characterized by three parameters $g$, $d$, and $t$, where $g$ is the number of groups, $d$ is number of days a group is continuously scheduled, and $t$ is the gap between the schedules. We call this a *(g, d, t) schedule*. The normal five day work week schedule is (1, 5, 2), where the entire workforce is just one group scheduled continuously for five days with two days off. A quintessential group schedule example that we highlight is the (2, 5, 0) schedule ((2, 4, 0) also has quite similar performance—see below). This schedule partitions individuals into two groups scheduled alternatively for five days each without any break between cycles. Thus, individuals cycle between five days of work and five days (quarantined) at home.

We evaluate our mechanisms in two main aspects:

1. How disease spreads under the mechanism and comparing it with simple baseline mechanisms. In particular, we posit the so-called *flattening ratio* which is the ratio of the peak number of cases (during the course of the epidemic simulation) under our mechanism compared to the peak number of cases under a baseline mechanism, e.g., the normal 5-day work week.

2. The in-person activity of the mechanism characterized by what we call the *in-person work ratio* or simply *work ratio* defined as the *ratio* of the average number of in-person working hours of an individual under our mechanism to the average number of working hours under the standard (pre-Covid) five-day working week (i.e., the (1, 5, 2) schedule). We assume that an individual works for a fixed number of in-person hours (say, 8) on each day they are working. For a mechanism with parameters ($g$, $d$, $t$), the work ratio is computed by the formula (see S1 Appendix for a derivation):

$$\frac{7}{5}\left(\frac{d}{gd+t}\right)$$

The work ratio for a normal work week, i.e., the (1, 5, 2) schedule, is 100%, whereas for the (2, 4, 0) (and (2, 5, 0)) schedule it is 70%.

There are two advantages in group scheduling: (1) the *average* number of contacts per group is reduced by a factor of $1/g$ (compared to a single group) which means that fewer individuals are infected by an infected person on average per day; (2) since infected symptomatic individuals in a group are quarantined when they are not scheduled, the number of days a person is infectious is reduced. Thus even when the number of groups is relatively small (say 2, 3, or 4) and even for small $d$ and $t$ values, the spread of disease is significantly reduced, while still maintaining a reasonable work ratio.

We demonstrate the efficacy of our mechanisms both theoretically (cf. Section 2.3.1) and experimentally (cf. Section 2.3.2). For our theoretical analysis, we consider a simple branching process model and analyze how the disease spreads as a function of the mechanism parameters ($g$, $d$, and $t$) and the COVID-19 disease parameters. Our analysis is fruitful in determining which mechanisms are likely to work well and also provides *insight* into our simulation results on various epidemiological models, which we discuss next.

## 2.2 Transmission model and disease parameters

Infectious diseases such as COVID-19 spread by contact between people. While various factors influence the spread of diseases, including COVID-19, we focus on a key ingredient that contributes to the spread of disease: the number of contacts between people and the distribution of the number of contacts (some people may have significantly more contacts than average). We simulate our mechanisms under two different types of standard epidemiological models: network-based, random-mixing based [4]. We use real-world data in our epidemiological models. Network-based models use an underlying *fixed graph structure* that determines the disease propagation, while random mixing models such as the traditional differential equations-based SIR (or SEIR) allow contacts between random individuals in the population (though the average number of contacts might be the same in both models). *We show that our mechanisms gives qualitatively similar benefits regardless of the specific models used in our simulations as well as the specific choice of parameters of the respective models.* These are discussed in detail in Section 2.3.2.

**2.2.1 Network-based contact models.** We use a simple graph-based model that is based on *contact distribution from real-world data* (Mossong et al. [5]). The work of Mossong et al.

[5] was a fairly large-scale (involving 7290 individuals and 92,904 contacts), population-based survey of "epidemiologically relevant" social contact patterns.

We define a *contact graph* where nodes consist of individuals and edges (assumed to be undirected) denote contacts between them. To model spatial distribution, we assume that nodes are distributed randomly in a unit square. This is a variant of the *random geometric* graph model that has been extensively studied [6]. In this random geometric graph model that we call the $G(n, k)$ model, we have $n$ nodes uniformly distributed in a unit area and each node has an edge with its $k$ closest neighbors. Note that this model has two key features: (1) spatial locality—edges are between nodes that are in proximity (2) the small-world property—neighbors of a node are themselves likely to be connected.

One might point out that having a uniform distribution of nodes (people) is not really reflective of the real world. While true, we posit that this is of lesser importance, especially when restricted to modelling densely populated areas such as Manhattan or Dharavi (Mumbai), or a university or a industrial workplace. However, more important is the modelling of the *contacts between people*. (In any case, as mentioned earlier, it is important to point out that the efficacy of the mechanisms are qualitatively similar regardless of the models, in particular whether it is contact-graph based or random mixing-based. Even in contact-based model, we consider models where geometry is less important, as discussed later. We also get similar results when modelling based on real-world contact data from Kissler et al. [13], see 2.2.4.) Hence, instead of adding an edge between a node and its $k$ closest contacts, where $k$ is fixed for all nodes, we use the well-studied real-world data for contact distribution due to Mossong et al. [5] to sample the number of contacts $k(v)$ for each node $v$. For a node $v$, $k(v)$ is its degree in the contact graph and its neighbors constitute the set of individuals that $v$ can infect directly. The work of Mossong et al. [5] studied the number of contacts for over 7000 people across eight countries in Europe. This data gives a contact distribution for the number of contacts of each node *per day*. The *mean number of contacts* for a person per day, according to this distribution, is 13.4, which we denote by $C$. We note that this contact distribution is for "normal" human behavior (i.e., no social distancing, quarantining, etc.).

We also consider an alternative contact graph model that is based solely on the contact distribution and *ignores the underlying geometry*. We call this $G'(n, D)$ model. It is a variant of a well-studied random graph model called the *Chung-Lu model* that is based on degree distribution [7]. Formally, the $G'(n, D)$ model (which is also undirected) is defined as follows. For each node $v$, we sample the (expected) degree of $v$, $d(v)$, from the contact distribution $D$. We then construct a *random graph* as follows. For each pair of nodes $u$ and $v$, an edge is added between $u$ and $v$ independently with probability $d(u)d(v)/\sum_u d(u)$. Note that, under this model, the expected degree of $v$ is equal to $d(v)$. We note that one important difference between the random geometric graph model and the random graph model is that the diameter of the Chung-Lu model is substantially smaller (about logarithmic in the size of the graph) than the random geometric graph. This means that the disease can potentially spread faster among the population, since there are shorter paths between nodes.

Finally, we also consider a parameterized hybrid model that interpolates between the random geometric model and the Chung-Lu random graph model, depending on a parameter $p$. In this model, each node has a degree (say $d$) randomly chosen from a distribution. The $d$ neighbors are then chosen as follows—with probability $p$, the *next closest neighbor* (in the geometric sense) is chosen as a neighbor of the node. With probability $1-p$, the neighbor is chosen randomly from *all nodes* in a weighted distribution with weights proportional to the degree of each node (in the Chung-Lu sense).

**2.2.2 COVID-19 disease model.** We now discuss how we model the spread of disease on the underlying contact graph. We employ a parameter called the *transmission probability*, $T_p$,

the probability that an infected node infects each of its uninfected contacts (neighbors in the contact graph) on *any given day*. Note that the $T_p$ value is related to the commonly used *reproductive rate* (or *effective reproductive rate*) $R(t)$ which measures (on average) the number of individuals an infected individual infects over the course of his/her infection (at any particular time $t$ during the epidemic). For the contact-graph model, one can approximately relate $T_p$ and $R(t)$ as follows (See S1 Appendix for a derivation):

$$R(t) = \left(1 - (1 - T_p)^D\right) \times C$$

where *C*, as defined earlier, is the average number of contacts per person (we assume this value to be 13.4 throughout this paper based on Mossong et al. [5] data) and *D* is the average number of days a person remains infectious (we assume in this paper that this number is 11 days for COVID-19 [8]). Note that in the contact graph model, since the underlying graph structure is fixed, $R(t)$ generally cannot exceed *C*. For a random mixing model, the relationship between $T_p$ and $R(t)$ is somewhat different (cf. Section 2.2). For example, currently for Houston, the reproductive rate is around 1 and thus the $T_p$ value under the above model is (approximately) 0.007. Generally, the reproductive rate of COVID-19 is estimated to be less than 3 [8] which corresponds to $T_p = 0.022$ (approximately) in the above contact graph model.

While the traditional approach in epidemiological modelling is to predict disease spread by estimating $T_p$ or $R(t)$ values and fitting these estimates in a model [9–11], we do not estimate these values. Rather, we study our mechanisms under various possible values for $T_p$ and compare the effectiveness of our mechanisms under different possible values with the baseline mechanisms discussed earlier. A high $T_p$ value means that the disease is spreading rampantly, while a low value means that the disease is spreading relatively slowly. We show that regardless of the value of $T_p$, our specific group scheduling mechanisms significantly reduce the infection spread compared to the baseline mechanisms. However, the efficacy increases even further as the $T_p$ values decrease (cf. Section 2.3.2).

For modeling the disease progress, we adopt an *SEIR model*, where individuals can be in four categories. Initially, all individuals are considered *Susceptible* to the disease. When a susceptible individual becomes infected, they first enter an *Exposed* state, wherein the individual is not contagious. After a period of time, the individual then enters the *Infected* state. During the infected state, an individual can transmit the disease with probability $T_p$ per contact per day. Then, after a certain period (called the *recovery time*), the individual becomes *Removed* (i.e., either recovered or deceased). We assume that the percentage of "deceased" individuals is very small compared to the population and does not affect the disease spread significantly; hence we ignore this in our simulation and assume that all individuals eventually recover. A removed individual cannot spread the infection to its neighbors.

We start with a set of randomly chosen individuals—called the *index set*—infected at the beginning of the simulation. We assume that the size of the index set is proportional to the *current* size of infected individuals in the population (e.g., in Harris County, Texas, it is about 3% in August 2020 [12]).

The *Incubation period* is the time between becoming infected and the development of symptoms. On average, it is estimated that, for COVID-19, it is 5.1 days, but can vary from 2-14 days [3, 8]. For our simulations, we use a distribution-based model for incubation period for COVID-19 [3]: we assume that the incubation period is given by a lognormal distribution with mean (approximately) 5 (days).

We assume that an infected person becomes contagious 2 days before the incubation period [8]. After a person becomes contagious, they can infect each of their contacts with probability $T_p$ per day. We consider various values of $T_p$ ranging from very high (say, 0.5) to relatively low

(say, 0.01)—as mentioned earlier, one can directly relate $T_p$ to the reproductive rate. After 14 days recovery time, an individual is Removed (i.e., either recovered or deceased). A removed individual is assumed to not be contagious.

It is known that *asymptomatic* carriers of COVID-19 play a significant role in the spread of the disease. It is estimated that as many as *40% of infected individuals are asymptomatic*, i.e., they do not show any symptoms, but continue to infect their contacts until they become Removed. We assume a similar estimate in our analysis, i.e., we assume each infected individual is asymptomatic with probability 0.4 (independently of others). We assume that asymptomatic carriers are as infectious as symptomatic carriers and become infectious in a similar time frame (i.e., two days before the incubation period, though they do not show symptoms).

**2.2.3 Random mixing model.** To study the robustness of our mechanisms across different models, besides the contact graph models and its variants described earlier, we also consider a classical SEIR random mixing model. The SEIR model tracks stages of a disease—susceptible, exposed, infected, and recovered—as the number of individuals in each stage. The evolution of each compartment is regulated by standard differential equations (see e.g. Keeling et al. [4]). We use a variation of the classical SEIR model with a lognormal distribution for the incubation period. The random mixing model allow each individual to have contact with random individuals which enables the possibility of larger values for the reproductive rate $R(t)$, since any susceptible individual can become exposed. By contrast, the contact graph model limits the susceptible population to the subset of nodes adjacent to infectious carriers. One can relate $R(t)$ and $T_p$ as: $R(t) = T_p \times C \times D$.

**2.2.4 Real world data model.** Finally, to provide some real-world evidence of the strength of our mechanisms, we utilize the results from Kissler et al. [13], specifically the Haslemere dataset of pairwise distances between volunteers over time. This dataset has already been used in modeling COVID-19 [14].

The Haslemere dataset "consists of the pairwise distances of up to 1m resolution between 469 volunteers from Haslemere, England, at five-minute intervals over three consecutive days (Thursday 12 Oct – Saturday 14 Oct, 2017)" [13]. We use this data set in our simulations to construct a contact graph wherein an edge exists between two participants if, at any time, they are a distance apart of 5 meters or less.

## 2.3 Analysis of scheduling mechanisms

**2.3.1 Theoretical analysis of mechanisms.** We present a theoretical analysis assuming a simplified branching process model which is much easier to analyze than the contact graph models defined earlier. The simplified model ignores the underlying graph model yet it gives useful insights to the efficacy of various scheduling mechanisms which are also validated by extensive experimental simulations on the various network models.

A main goal of our analysis is to study the efficacy of mechanisms with respect to disease spread. In particular, given a value of transmission rate $T_p$, we would like to discover mechanisms (with good work ratio) that can effectively control the spread of disease. The evolution of the disease depends on the transmission rate $T_p$ (with greater values of $T_p$ corresponding to a greater rate of spread), the number of contacts per individual, the number of days a person is infectious, and the asymptomatic rate. It is important to note that our mechanisms employ symptomatic quarantine and hence symptomatic individuals contribute less to the spread (since they are infectious for a short while before symptoms appear, after which they become quarantined). However, asymptomatic carriers (who constitute about 40% of infected individuals) play a major role in spreading the disease. We analyze how the disease spreads under a given $(g, d, t)$ mechanism.

Our analysis uses the well-known Galton-Watson branching process (e.g., see Section 8.1 [15]) to study disease evolution. In a branching process, each (infected) individual independently infects an $X$ number of individuals, where $X$ is a random variable (capturing the reproductive rate) with a fixed distribution (with finite mean and variance).

In a branching process analysis, we start with an infected individual and study how the number of infected individuals grow in each generation (see Fig 1). Table 1 lists the key parameters (and their typical values) used in our analysis.

Consider an infected individual, say node $v$. By our modeling assumptions, $v$ will be symptomatic with probability 0.6 and asymptomatic with probability 0.4. Let $\mu_s$ (resp. $\mu_a$) be the expected number of people that are infected by a symptomatic (resp. asymptomatic) infected individual. Essentially, $\mu_s$ and $\mu_a$ are the effective reproductive rates $R(t)$ for the symptomatic and asymptomatic patient respectively. Let $X_i$ be the random variable counting the number of infected people at the $i^{th}$ generation and $\mathbb{E}[X_i]$ be the expected value of $X_i$. Assuming $X_0 = 1$, $\mathbb{E}[X_1] = 0.6\mu_s + 0.4\mu_a$, and in general, $\mathbb{E}[X_i] = (0.6\mu_s + 0.4\mu_a)^i$, which follows from the branching process, See Fig 1 (a formal argument can be found in van Handel [15]).

Thus, the expected total number of infected individuals is $\sum_{i=1}^{\infty} (0.6\mu_s + 0.4\mu_a)^i$ is the sum of a geometric series with geometric mean $r = 0.6\mu_s + 0.4\mu_a$. Therefore, if $(0.6\mu_s + 0.4\mu_a) < 1$, i.e., each individual infects less than one person on average, then $\mathbb{E}[X_i] \to 0$ as $i \to \infty$. In this case, the disease dies out eventually. On the other hand, if $0.6\mu_s + 0.4\mu_a > 1$ (i.e., each individual infects more than one person on average), then $\mathbb{E}[X_i]$ grows exponentially. In this case, the disease-spread explodes and ultimately infects the entire population. Thus, we have the following theorem.

**Theorem 1**. *If $0.6\mu_s + 0.4\mu_a < 1$ then the branching process dies out eventually, i.e., the disease eventually stops spreading.*

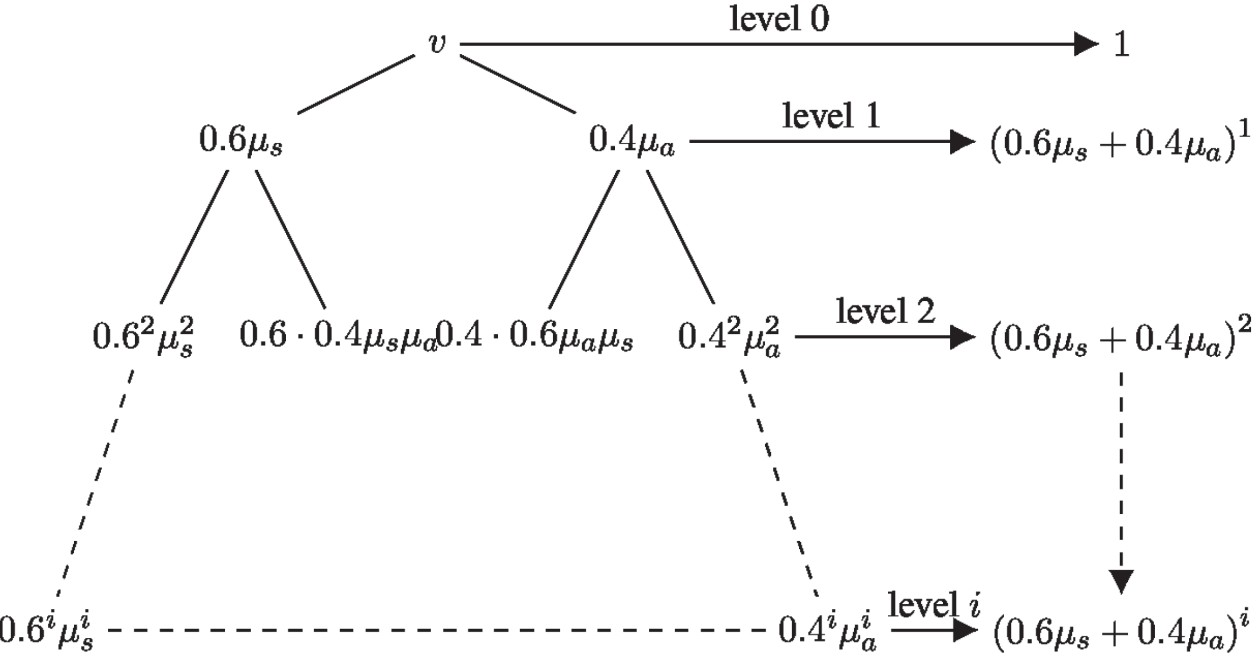

**Fig 1. Illustrating the branching process analysis.** The right hand side displays the total expected number of infected people at each level.

**Table 1. COVID-19 parameters used in our analysis.**

| Parameter | Definition | Value Used in our Analysis |
|---|---|---|
| $T_p$ | probability that a contagious person infects a single neighbor in a day | {0.01, 0.1} |
| asymp | probability that an infected person is asymptomatic | 0.4 |
| incubation period | number of days until symptoms develops from infection | 5 |
| $C$ | number of contacts per person | 13.4 |
| $D$ | number of days an infected person remains infectious | 11 |

Below we calculate the value of effective reproductive rate $0.6\mu_s + 0.4\mu_a$ in general for a mechanism with parameters $(g, d, t)$. Then we analyze different specific mechanisms (i.e., with specific values for $g$, $d$, and $t$) to check whether the disease dies out or not under the mechanism.

In the random mixing model, the reproductive rate is $R(t) = T_p \times D \times C$, where $T_p$ is the transmission probability, $D$ is number of days a person remains infectious and $C$ is the number of contacts per person. By considering a schedule $(g, d, t)$, the above formula becomes:

$$R(t) = T_p \times D_g \times C_g, \tag{1}$$

where $C_g = C/g$ is the number of contacts in a random grouping of $g$ groups and $D_g$ is the number of days a person is infectious as determined by the schedule (note that if a group is not scheduled, then any person in that group is not considered infectious, even though they may be infected). The following lemmas show how we can compute the symptomatic and asymptomatic reproductive rates for a specific $(g, d, t)$ mechanism. The proofs (see Appendix A.4 in S1 Appendix) are derived from Eq 1 with appropriate calculation of $D_g$ and $C_g$.

**Lemma 1**. *Let* $d' = \max\{0, \min\{d - 2, 5\}\}$ *and* $d_i' = \max\{0, \min\{5 - (gd + t)i, d\}\}$. *Then the Symptomatic reproductive rate is given by*: $\mu_s = \frac{(13.4)T_p}{g} \cdot \left(d' + \sum_i d_i'\right)$

**Lemma 2**. *Let* $d'' = \max\{0, \min\{d - 2, 11\}\}$ *and* $d_i'' = \max\{0, \min\{11 - (gd + t)i, d\}\}$. *Then our Asymptomatic reproductive rate is given by*: $\mu_a = \frac{(13.4)T_p}{g} \cdot \left(d'' + \sum_i d_i''\right)$

Thus, one can calculate $0.6\mu_s + 0.4\mu_a$ from the above formulas. Table 2 demonstrates these valuse for different mechanisms for two different $T_p$ values—0.1 (high) and 0.01 (low). If the value of $0.6\mu_s + 0.4\mu_a < 1$ for a mechanism and a $T_p$ value, then the disease dies out; moreover, the closer this value is to 0, the faster the disease dies out (and ends up infecting a smaller fraction of the population). These predictions are validated by simulation results in the various models described in Section 2.2—network-based, random mixing, and real-world network (Section 2.3.2).

**2.3.2 Simulation results.** To study the efficacy of our mechanisms, we also conduct extensive simulations across various models and parameters [16]. For example, given a particular

**Table 2. Values of $0.6\mu_s + 0.4\mu_a$ for different values of $T_p$ and schedules $(g, d, t)$.**

| $T_p$ | Schedule | | | |
|---|---|---|---|---|
| | (1, 5, 2) | (2, 5, 0) | (3, 3, 0) | (4, 4, 0) |
| 0.01 | 0.616 | 0.228 | 0.080 | 0.067 |
| 0.10 | 6.164 | 2.278 | 0.804 | 0.670 |

A green box indicates that the process eventually dies out, whereas a red box indicates continued growth.

mechanism, say (2, 4, 0), we simulate the disease spread under this mechanism under various models (the contact graph model and its variants, the random mixing model, and the Haslemere real-world data model). Under each model, we vary the following parameters to study their effects. We vary transmission probability $T_p$ (which captures the rate of infection spread in the population), the number of index patients (which captures the percentage of individuals currently infected among the population), and the percentage of asymptomatic carriers (we assume this to be 40% according to current estimates [2], but also simulate with other values). We then compare the disease spread, under the *same set of respective parameters*, to three baseline mechanisms—the basic model (where the disease spreads without any intervention), symptomatic quarantine or the (1, 1, 0) schedule (where infected individuals are quarantined after exhibiting symptoms), and the (1, 5, 2) schedule, which is the normal 5-day work week with symptomatic quarantine (note that in the latter two mechanisms there is only one group). The key metric of comparison is the flattening ratio—the ratio of the peak number of cases of the mechanism under consideration (say (2, 4, 0)) to that of the baseline mechanisms. We also compare the total number of infections.

We analyze a number of group scheduling mechanisms that showcase the trade off between the work ratio (WR) and the disease spread. We categorize them into three broad groups as: (i) high-WR mechanisms ($\sim 70\%$ work ratio), (ii) mid-WR mechanisms ($40 - 50\%$), and (iii) low-WR mechanisms (about 30%).

Our results are summarized in Figs 2 to 4 and Table 3. (Figs 3 and 4 are placed in the appendix). These particular results assume a population of 50,000 (the typical population in a large university) and the number of index patients is 3% of the population (i.e., 3% of the population is initially infected as estimated, say, in Harris County, TX in August 2020 [12]). The simulation model is the random geometric model (Fig 2) with the contact distribution as described in Section 2.2, for the random mixing model (Fig 3), and for the Real-World Data model (Fig 4). We adopt the COVID-19 disease parameters as described in Section 2.2. *Our results are qualitatively similar across the various models including other variants of the contact graph model, the random mixing model, and the real-world data model.* We have simulated higher populations (up to 100,000 in the contact graph model, and up to million in the random mixing model) and have varied the number of index patients. More importantly, we have analyzed a wide variety of $T_p$ values—here we consider two canonical $T_p$ values—0.1 and 0.01—these capture high and low reproductive rates respectively (cf. Section 2.2).

Our results can be summarized as follows (in particular, see Fig 2 and Table 3). The canonical example for the high-WR category is the (2, 5, 0) schedule, which achieves a (1, 5, 2) flattening ratio as low as 12% (i.e., the ratio of the peak number of cases is 12% compared to that of the standard work week schedule) even when $T_p = 0.1$. When $T_p$ is lower, say around 0.01 the flattening ratio becomes much lower. (In general, the lower the $T_p$, the better the flattening, in general, for any given mechanism.) In fact, several mechanisms of the form (2, $d$, 0) yield the same work ratio and essentially the same flattening ratio for $d \geq 4$. In the mid-WR category, we get even better flattening ratios. The canonical example for this case is (3, 3, 0) yielding a work ratio of 46%, but with the flattening ratio down to 4% even under $T_p = 0.1$. For mechanisms in low-WR category, such as (4, 4, 0) with still a reasonable work ratio of 35%, the flattening is down to about 1%. The mid-WR and low-WR mechanisms are attractive when $T_p$ values are high, as they not only lead to a low flattening ration, but also low number of peak cases (in absolute numbers) with respect to the total population. Moreover, the total number of infections is a small fraction of the total population.

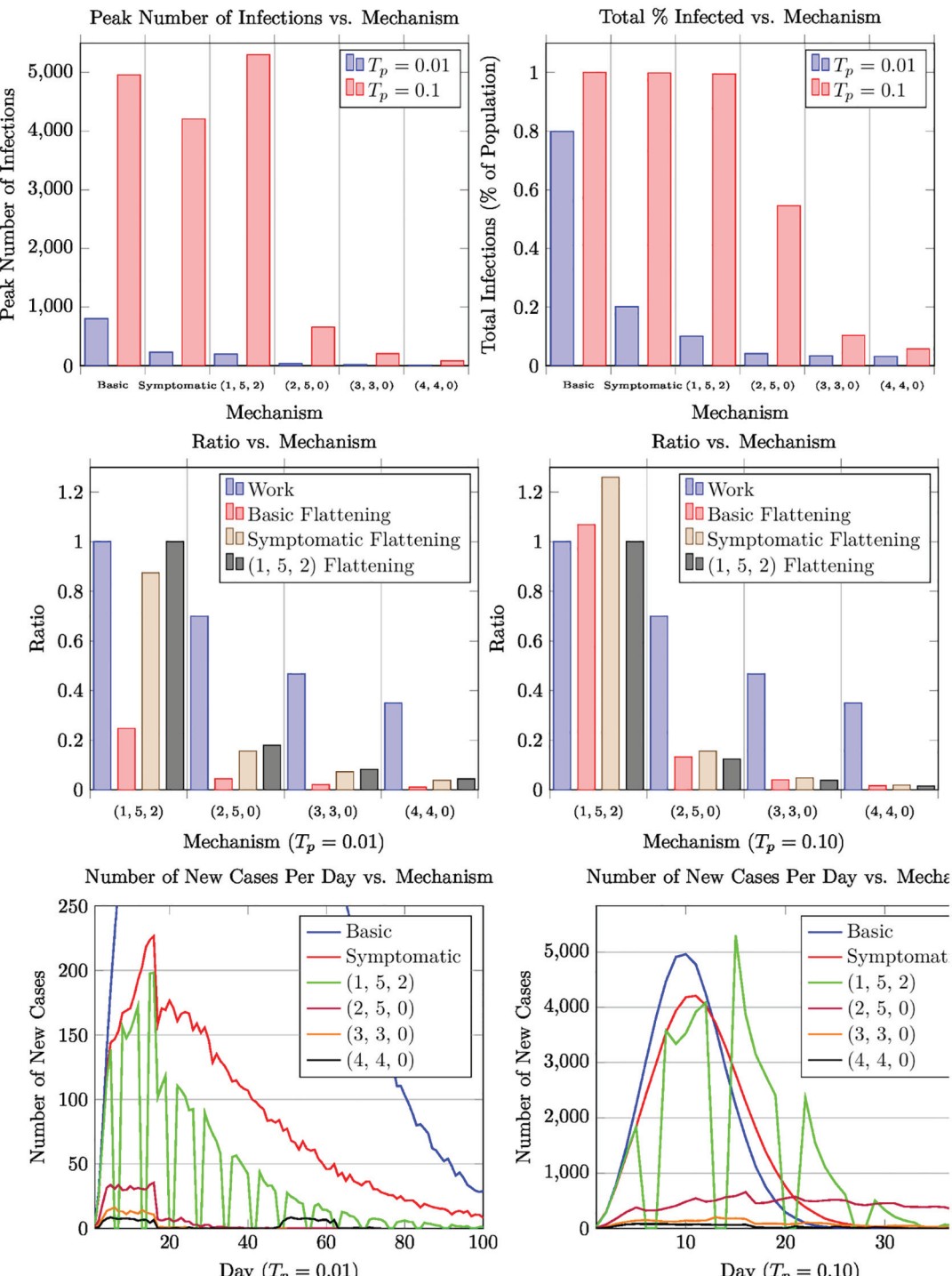

**Fig 2. Plots displaying the performance of different mechanisms for Contact Graph Model.**

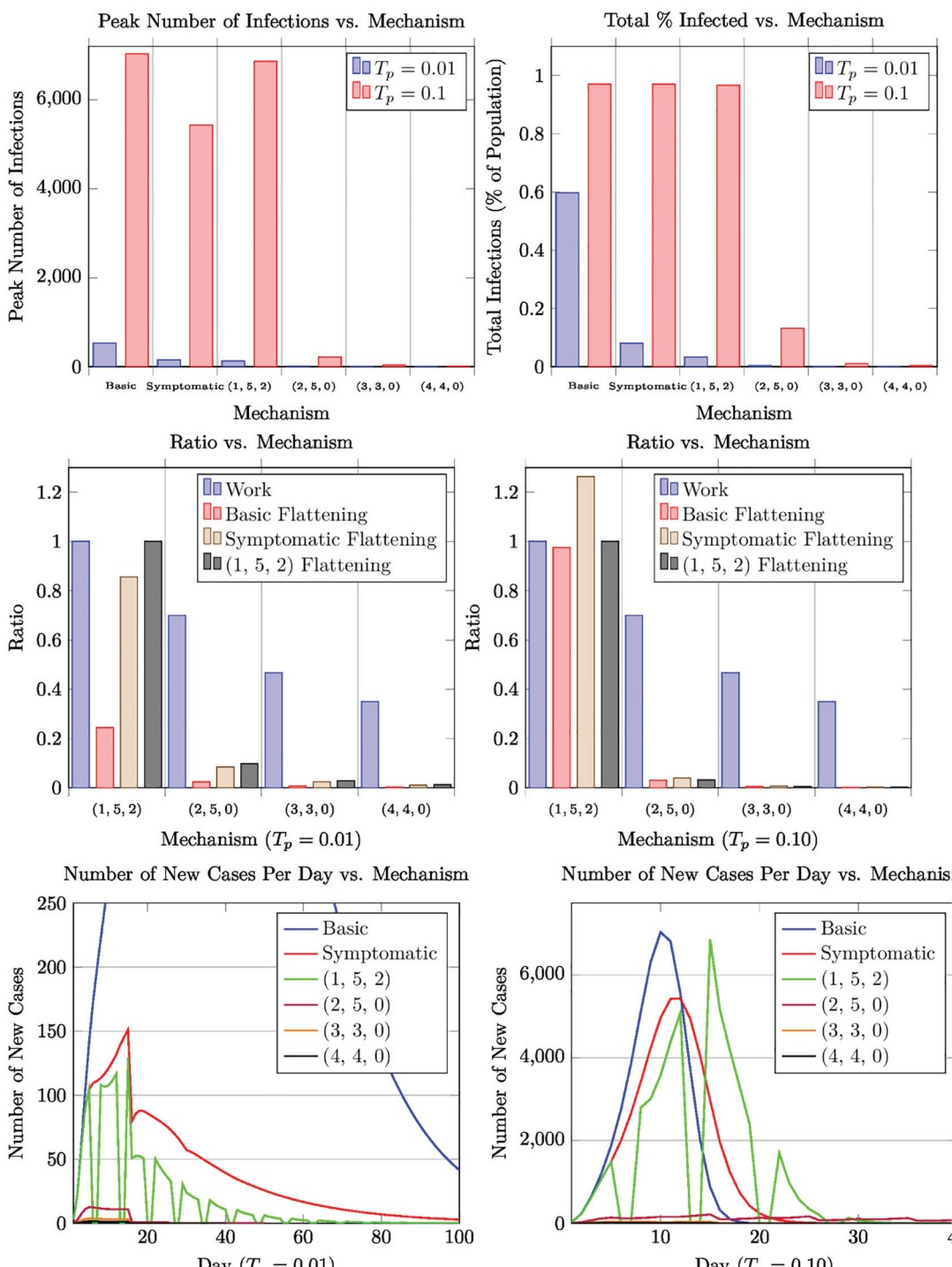

**Fig 3. Plots displaying the performance of different mechanisms for the random mixing Model.**

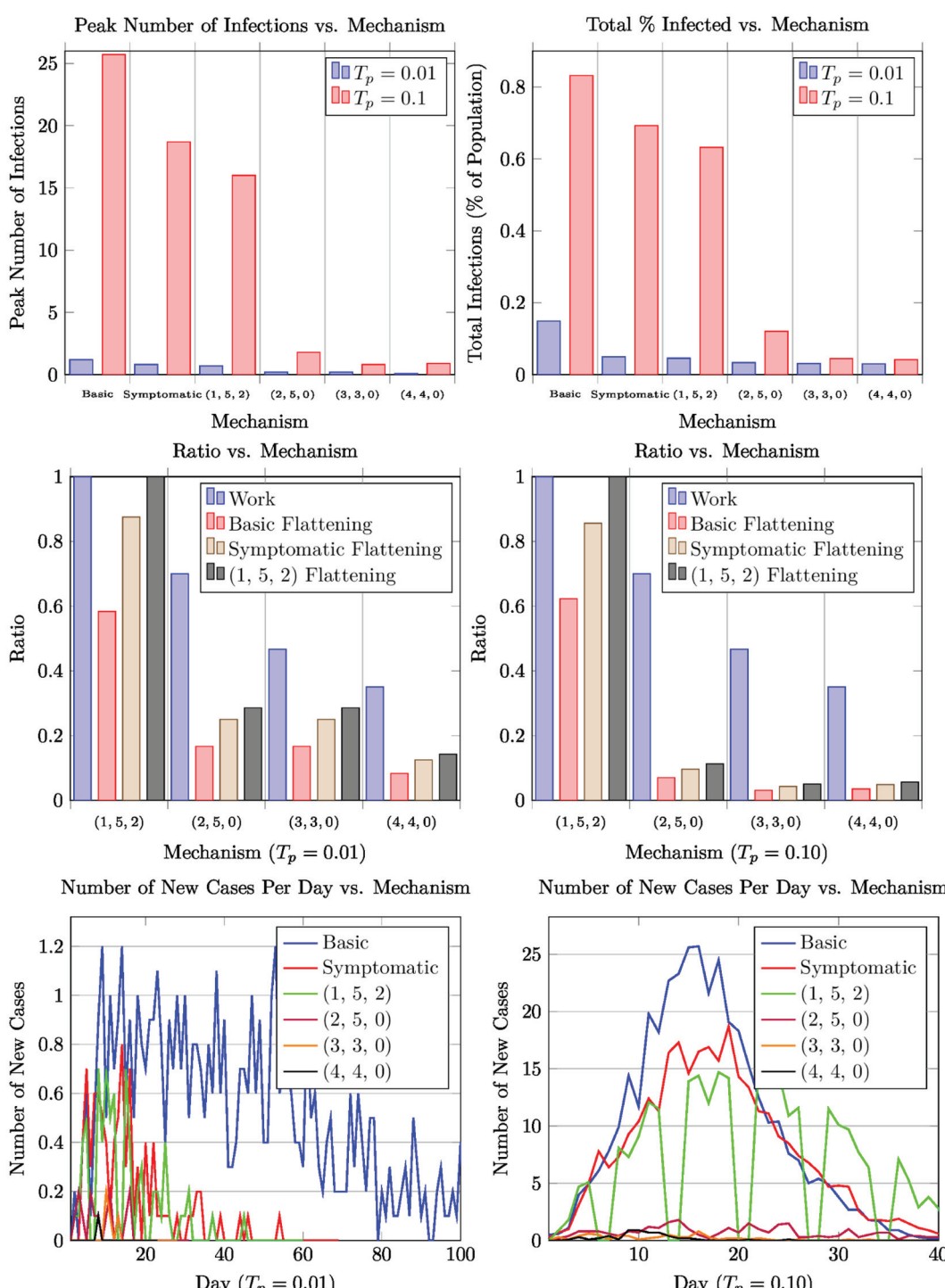

**Fig 4. Plots displaying the performance of different mechanisms for modified Haslemere dataset.**

**Table 3. Work and flattening ratios for various schedules against the basic, Symptomatic Quarantine, and (1, 5, 2) mechanisms for $T_p$ values of 0.01 and 0.1 in the contact graph model.**

| WR Category | Schedule | WR | $T_p = 0.01$ | | | $T_p = 0.1$ | | |
|---|---|---|---|---|---|---|---|---|
| | | | Basic | Sympt. | (1, 5, 2) | Basic | Sympt. | (1, 5, 2) |
| Full | (1, 5, 2) | 100% | 23% | 80% | | 107% | 127% | |
| | (1, 1, 0) | 140% | 29% | | 124% | 84% | | 79% |
| High | (2, 4, 0) | 70% | 4% | 14% | 18% | 14% | 17% | 13% |
| | (2, 5, 0) | 70% | 4% | 14% | 18% | 12% | 15% | 12% |
| Mid | (2, 3, 2) | 53% | 4% | 15% | 18% | 11% | 14% | 11% |
| | (2, 3, 3) | 46.7% | 4% | 14% | 18% | 11% | 13% | 10% |
| | (3, 3, 0) | 46.7% | 2% | 7% | 8% | 4% | 5% | 4% |
| Low | (4, 4, 0) | 35% | 1% | 3% | 4% | 2% | 2% | 1% |

A higher work ratio indicates more in-person activity, whereas a lower flattening ratio indicates a lower peak number of new cases per day.

## 2.4 Related work

The results on epidemiological studies is vast and here we focus on the literature that is most relevant to our work on Covid-19 and similar strategies.

Our work closely resembles several recent papers on cyclical strategies [17–19] and alternating strategies [20]. A typical cyclical strategy parameterized by $1 \leq k < 14$ views 14 days as one work cycle and stipulates that the population works for $k$ days followed by $14 − k$ days of lockdown. Our work is complementary to their works. The key difference is that group scheduling mechanisms benefit significantly from the reduced number of contacts owing to the partitioning of the population whereas the cyclical mechanisms typically do not. Moreover, our approach to modeling in-person activity is much simpler than the works by Alon et al. and Ely et al. [18, 19], which model economic impact, and is quite likely to be more intuitive and easier for policy makers to reason about and compare alternatives.

A number of works study specific mechanisms that resemble our work, but few provide the trade off analysis between productivity and infection spread that is crucial for policymakers. For example, Karin et al. [17] briefly discusses a staggered form of cyclical strategies. In Maiden et al. [20], the authors focus their efforts on an alternating quarantine mechanism whereby they partition the population into two groups and schedule activities alternating each week between the two groups. Being primarily focused either on typical cyclical strategies [17] or alternating quarantine [20], neither of these works analyze the trade off with respect to productivity and the rate at which the disease spreads. Alon et al. [18] provides a detailed model that studies the trade off between the effectiveness of typical cyclical strategies and the economy. However, we believe that group scheduling mechanisms can provide significantly better trade off. There has been a flurry of recent works [21–23] that study other mechanisms like social distancing and targeted lockdowns in the context of their related economic impact.

## 2.5 Discussion

We studied group scheduling mechanisms and demonstrated their efficacy both theoretically and experimentally. To gain theoretical insight, we considered a simple branching process model and analyzed how the disease spreads as a function of the mechanism parameters ($g$, $d$, and $t$) and the COVID-19 disease parameters. Our analysis is helpful in determining what mechanisms are likely to work well. We also conducted extensive simulations of our mechanisms under various epidemiological models and compare their performance with baseline mechanisms.

*We show that our mechanisms gives qualitatively similar benefits regardless of the specific models used in our simulations as well as the specific choice of parameters of the respective models.* For example, given a particular mechanism, say (2,4,0), we simulate the disease spread under this mechanism under various epidemiological models. Under each model, we vary the following parameters to study their effects. We vary transmission probability (which captures the rate of infection spread in the population), the number of index patients (which captures the percentage of individuals currently infected among the population), and the percentage of asymptomatic carriers (we assume this to be 40% according to current estimates, but we also try other values). We then compare the disease spread, under the *same set of respective parameters*, to three baseline mechanisms—the basic model (where the disease spreads without any intervention), symptomatic quarantine or the (1,1,0) schedule (where infected individuals are quarantined after exhibiting symptoms), and the (1,5,2) schedule, which is the normal 5-day work week with symptomatic quarantine (note that in the latter two mechanisms there is only one group). The key metric of comparison is the flattening ratio—the ratio of the peak number of cases of the mechanism under consideration (say (2,4,0)) to that of the baseline mechanisms. We also compare the total number of infections as well.

We analyze a number of group scheduling mechanisms that showcase the trade off between in-person work ratio and the disease spread. We categorize them into three broad groups as: 1. High-WR mechanisms (70% work ratio), 2. Mid-WR mechanisms (40–50%), and 3. Low-WR mechanisms (about 30%). The canonical example for the high-WR category is the (2, 5, 0) schedule, which achieves a (1,5,2) flattening ratio as low as 12%. The mid-WR and low-WR mechanisms are attractive when the transmission rates are high, as they not only lead to a low flattening ratio, but also low number of peak cases (in absolute numbers) with respect to the total population. Moreover, the total number of infections is a small fraction of the total population and the infection dies off quickly in the population.

Regarding the simulations, we point that the models that we use are to *compare* different mechanisms. To show the robustness of our results, we consider several models (and several settings of various parameters in each model) as well as a real-world network model (based on face-to-face interactions in a group of people) and in all the models we compare group scheduling mechanisms with other baseline mechanisms and show the benefit of group scheduling. For example, one of the models is the SEIR random mixing model, a standard model in epidemiology. We believe that although the numbers that we predict such as the flattening ratio may not exactly match in the real world, we believe that group scheduling will still yield significant benefit.

Our mechanisms take a principled approach to disease control that interpolates between extreme measures—lockdowns on one hand that severely cripple the economy and a "herd immunity" approach that advocates normal behavior for most people (except the most vulnerable). The latter approach, though it helps economic activity, has the danger that even younger or middle-aged people who apparently are less vulnerable can still get the disease in severe form (and even die), and this can happen in large numbers, overwhelming the health care system.

We note that the group scheduling mechanisms assume that the population is randomly partitioned into $g$ groups. Random partitioning is important to justify that the average number of contacts each person has will go down by a factor of $g$ on average. This kind of partitioning is more applicable to structured settings like schools and workplaces. Our focus is mainly on such settings (even in our simulation we assume about 50,000 people, a typical population in a large public university).

For convenience, we have categorized our mechanisms based on their work ratios. The low end of this spectrum provides the best flattening ratio. So naturally, when cases surge

(characterized by a higher transmission probability or reproductive rate) or when easing out of complete lockdown, we may wish to opt for the low work ratio options that have an improved flattening ratio. As case numbers decrease, we have a couple of strategies that we can choose between. On the one hand, low case numbers afford us the ability to contact trace more effectively and thereby stop the spread. On the other hand, we can also move to mechanisms with improved work ratio.

*An important point to note is that if the transmission probability (reproductive rate) is reduced, then the efficacy of group scheduling is increased even further.* Thus reducing the transmission probability by following public health guidelines like wearing masks, social distancing, and hand washing will be very beneficial. Moreover, following these guidelines can increases the efficacy of the mechanisms, allowing deployment of high-WR mechanisms as well. More recently, with increased availability of vaccines, most countries have initiated vaccination drives and have managed to vaccinate significant fractions of their population. Unfortunately, due to the slow production rate and other logistical issues, the extent of vaccination coverage varies widely across countries and regions within countries. Policymakers can factor in the reduced transmission rate based on the extent of vaccination coverage and consider operating at a higher work-ratio mechanism.

## 3 Conclusion

We are guided by two main principles. Firstly, our mechanisms—by partitioning the population into groups—reduce the number of contacts per person. Secondly, we schedule each group with sufficient break time so that a large number of infected people become symptomatic (and therefore quarantined) during their breaks. We believe that policy makers can be guided by these principles and adapt our mechanisms, taking their specific local considerations into account. In general, a policy *A* (that builds on another policy *B*) that decreases the number of contacts further or, improves the probability that people become symptomatic during their break, will dominate over *B*, and lead to better flattening. This means that adding common work-breaks—e.g., Sundays off—is likely to improve the flattening ratio and unlikely to worsen it.

Policy makers can also use the above two principles to address scenarios that we have not addressed directly. Shift workers, for example, may need to work on a more fine-grained schedule. Consider a shift schedule that requires two shifts per day. We could consider four groups, each alternating between four working days and four off days. The groups may be staggered so that—on any given day—two groups are working and two are off. Of particular interest is the application of this mechanism to health facilities and hospitals, where the mechanism can be used in scheduling the shifts of health workers, thereby reducing the tranmission of the virus [24].

Our model can be extended in many ways. We currently focus on social interactions at the workplace, but we believe our results will qualitatively extend even when we consider social interactions at home. Moreover, our current model does not explicitly consider age-related factors in terms of transmission rate and death rates. It is now quite well-established that younger people fare better on both counts. Thus, including these details into our model and scheduling mechanisms may help to further improve the trade-off between productivity and disease spread.

## Supporting information

**S1 Appendix.**
(PDF)

## Author Contributions

**Conceptualization:** John Augustine, Anisur Rahaman Molla, Gopal Pandurangan.

**Data curation:** Khalid Hourani, Gopal Pandurangan, Adi Pasic.

**Formal analysis:** John Augustine, Khalid Hourani, Anisur Rahaman Molla, Gopal Pandurangan.

**Investigation:** John Augustine, Khalid Hourani, Anisur Rahaman Molla, Gopal Pandurangan, Adi Pasic.

**Methodology:** Anisur Rahaman Molla.

**Project administration:** Gopal Pandurangan.

**Software:** Khalid Hourani.

**Supervision:** Gopal Pandurangan.

**Visualization:** Khalid Hourani.

**Writing – original draft:** John Augustine, Khalid Hourani, Anisur Rahaman Molla, Gopal Pandurangan.

**Writing – review & editing:** John Augustine, Khalid Hourani, Anisur Rahaman Molla, Gopal Pandurangan.

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
