## [Decision Letter · Decision Letter 0]

4 Apr 2022

PONE-D-21-33787Scheduling Mechanisms to Control Spread of Covid-19PLOS ONE

Dear Dr. Hourani,

Thank you for submitting your manuscript to PLOS ONE. After careful consideration, we feel that it has merit but does not fully meet PLOS ONE’s publication criteria as it currently stands. Therefore, we invite you to submit a revised version of the manuscript that addresses the points raised during the review process.

We look forward to receiving your revised manuscript.

Kind regards,

Maria Alessandra Ragusa, PhD Professor

Academic Editor

PLOS ONE

Journal Requirements:

" John Augustine’s research is supported in part by an Extra-Mural Research Grant (file number EMR/2016/003016) funded by the Science and Engineering Research Board, Department of Science and Technology, Government of India and by the VAJRA faculty program of the Government of India. Anisur Rahaman Molla’s research supported by DST Inspire Faculty research grant DST/INSPIRE/04/2015/002801. Gopal Pandurangan’s research is supported, in part, by NSF grants IIS-1633720, CCF-BSF-1717075, CCF1540512, US-Israel BSF award 2016419, and by the VAJRA faculty program of the Government of India."

"John Augustine's research is supported in part by an Extra-Mural Research Grant (file number EMR/2016/003016) funded by the Science and Engineering Research Board, Department of Science and Technology, Government of India and by the VAJRA faculty program of the Government of India.

Anisur Rahaman Molla's research supported by DST Inspire Faculty research grant DST/INSPIRE/04/2015/002801.

Gopal Pandurangan's research is supported, in part, by NSF grants  IIS-1633720,  CCF-BSF-1717075, CCF-1540512,  US-Israel BSF award 2016419, and by the VAJRA faculty program of the Government of India.

https://nsf.gov/

https://www.bsf.org.il/

https://www.vajra-india.in/

Additional Editor Comments:

To Corresponding Author,

I require before a possible acceptance that are followed all the modifications suggested by the referees.

So, I recommend you to read the reports and make the changes.

There are no conflicts between the review.

My decision of MINOR REVISION is justified on PLOS ONE journal according to the reports.

Best regards

Reviewers' comments:

Reviewer's Responses to Questions

**Comments to the Author**

1. Is the manuscript technically sound, and do the data support the conclusions?

Reviewer #1: Yes

Reviewer #2: Yes

2. Has the statistical analysis been performed appropriately and rigorously? 

Reviewer #1: Yes

Reviewer #2: Yes

3. Have the authors made all data underlying the findings in their manuscript fully available?

Reviewer #1: Yes

Reviewer #2: Yes

4. Is the manuscript presented in an intelligible fashion and written in standard English?

Reviewer #1: Yes

Reviewer #2: Yes

5. Review Comments to the Author

Reviewer #1: In this paper, the authors studied scheduling mechanisms that explore the trade-off between

containing the spread of COVID12 19 and performing in-person activity in organizations. The

proposed mechanisms are based on partitioning the population randomly 14 into groups and

scheduling each group on appropriate days with possible gaps. Finally, the authors demonstrated

the efficacy of our mechanisms by theoretical analysis and 24 extensive experimental simulations

on various epidemiological models based on real-world data.

The results involved in the work are new and through which several special cases can be concluded;

i.e. the approach and techniques can be modified in a straightforward manner to study the

spread of other diseases, including COVID-19 variants.

Furthermore, they analyzed a number of group scheduling mechanisms that showcase the trade

off between in-person work ratio and the disease spread.

Finally, I strongly recommend the publication of this paper in PLOS ONE.

Reviewer #2: Nowadays, the work is very interesting. During winter the circulation of the virus is beginning to be higher with the consequences on health. The ongoing COVID-19 pandemic produced an unprecedented health and economic crisis, urging for the development of adapted actions aimed to monitoring the spread of the new coronavirus. A new variant is spreading and the danger of a possible lock down have to be considered in the coming days.

The proposed model of group scheduling has the aim to maintain the activity in presence and operating a control of the spread of the virus. This model is a useful idea for the rulers.

In the text it is well illustrated which are the advantages in group scheduling even if the groups are few and small. Monitoring the evolution with the time of reproductive rate constitutes a critical factor in situations such as that of COVID-19, when decisions need to be taken and action need to be made under emergency.  Referring to the epidemiological model to predict disease spread by estimating Tp o Rt we can cite (A, B, C).  

In the real world, it may be useful to highlight the possible different effects of the group scheduling proposed to avoid contagion in different situations whereas the usual cyclical mechanisms typically do not. In the transmission of the virus must also be considered the possible influences determined by the behaviours more or less correct observed by the subjects. The authors have shown that Model 2.5.0 compared to Model 1.5.2, drastically reduces the number of peak infections and significantly the total number of infections in the population, but results in a 30% reduction in the work ratio. The proposed scheme would be able to achieve the result of reducing the transmission of the virus and reducing the contagion in health facilities or hospitals, even where the behaviours have not substantially changed a cause of pandemic (D). 

This is why the model could usefully be proposed in the construction of shifts of health workers. In the healthcare facilities, bearing the alternation provided for in the group scheduling mechanism presupposes a workforce that is numerically adequate to the standard needs. The latter could, with smaller health workers’ numbers, in the event of a pandemic, cope with a greater activity with sacrifice but would allow to continue to safeguard health and provide essential services. 

The identification of the three parameters for the identification of the group scheduling mechanism is clear. In my opinion, Table 1 does not add anything to what is described in the text. I would suggest to create a “Glossary of abbreviations” containing, in addition to the parameters used in the analysis, also the other acronyms and abbreviations used in the text for easier reading (g,d,t, n, k, v, u, R, X, m...etc.)

Suggested references

A.Abry P, Pustelnik N, Roux S, Jensen P, Flandrin P, Gribonval R, Lucas CG, Guichard É, Borgnat P, Garnier N. Spatial and temporal regularization to estimate COVID-19 reproduction number R(t): Promoting piecewise smoothness via convex optimization. PLoS One. 2020 Aug 20;15(8):e0237901. doi: 10.1371/journal.pone.0237901.

B.Ponjavić M, Karabegović A, Ferhatbegović E, Tahirović E, Uzunović S, Travar M, Pilav A, Mulić M, Karakaš S, Avdić N, Mulabdić Z, Pavić G, Bičo M, Vasilj I, Mamić D, Hukić M. Spatio-temporal data visualization for monitoring of control measures in the prevention of the spread of COVID-19 in Bosnia and Herzegovina. Med Glas (Zenica). 2020 Aug 1;17(2):265-274. doi: 10.17392/1215-20.

C.Hong HG, Li Y. Estimation of time-varying reproduction numbers underlying epidemiological processes: A new statistical tool for the COVID-19 pandemic. PLoS One. 2020 Jul 21;15(7):e0236464. doi: 10.1371/journal.pone.0236464.

D.Ragusa R, Marranzano M, Lombardo A, Quattrocchi R, Bellia MA, Lupo L. Has the COVID 19 Virus Changed Adherence to Hand Washing among Healthcare Workers? Behav Sci (Basel). 2021 Apr 15;11(4):53. doi: 10.3390/bs11040053.

6. PLOS authors have the option to publish the peer review history of their article (what does this mean?). If published, this will include your full peer review and any attached files.

Reviewer #1: **Yes: **Mohamed I. Abbas

Reviewer #2: No

---

## [Author Response · Author response to Decision Letter 0]

24 May 2022

Editor comments:

1. The manuscript and file names have been updated to follow the style requirements of PLOS ONE.

2. The funding information provided in the submission is correct. Additionally, we have written it in the Cover Letter.

3. We have removed the funding information from the acknowledgments. Additionally, we have written it in the Cover Letter.

Reviewer comments:

We thank the reviewers very much for their insightful comments. We also thank them for suggesting relevant citations and have added them to the manuscript. We thank the reviewers for the insight regarding health facilities and hospitals and have updated the manuscript accordingly. We have also added a glossary of abbreviations to the appendix.

---

## [Decision Letter · Decision Letter 1]

26 Jul 2022

Scheduling mechanisms to control the spread of COVID-19

PONE-D-21-33787R1

Dear Dr. Hourani,

We’re pleased to inform you that your manuscript has been judged scientifically suitable for publication and will be formally accepted for publication once it meets all outstanding technical requirements.

Kind regards,

Maria Alessandra Ragusa, PhD Professor

Academic Editor

PLOS ONE

Additional Editor Comments (optional):

The revised version is ready for publication. Best regards.

Reviewers' comments:

Reviewer's Responses to Questions

**Comments to the Author**

1. If the authors have adequately addressed your comments raised in a previous round of review and you feel that this manuscript is now acceptable for publication, you may indicate that here to bypass the “Comments to the Author” section, enter your conflict of interest statement in the “Confidential to Editor” section, and submit your "Accept" recommendation.

Reviewer #2: (No Response)

2. Is the manuscript technically sound, and do the data support the conclusions?

Reviewer #2: (No Response)

3. Has the statistical analysis been performed appropriately and rigorously? 

Reviewer #2: (No Response)

4. Have the authors made all data underlying the findings in their manuscript fully available?

Reviewer #2: (No Response)

5. Is the manuscript presented in an intelligible fashion and written in standard English?

Reviewer #2: (No Response)

6. Review Comments to the Author

Reviewer #2: The observations of the referees have been understood and the suggested changes to the text have been made.

7. PLOS authors have the option to publish the peer review history of their article (what does this mean?). If published, this will include your full peer review and any attached files.

Reviewer #2: No

---

## [Editor Report · Acceptance letter]

1 Aug 2022

PONE-D-21-33787R1 

Scheduling mechanisms to control the spread of COVID-19 

Dear Dr. Hourani:

I'm pleased to inform you that your manuscript has been deemed suitable for publication in PLOS ONE. Congratulations! Your manuscript is now with our production department. 

Kind regards, 

on behalf of

Dr. Maria Alessandra Ragusa 

Academic Editor

PLOS ONE